# To Bev or Not to Bev during Ovarian Cancer Maintenance Therapy?

**DOI:** 10.3390/cancers15112980

**Published:** 2023-05-30

**Authors:** Jacek Jan Sznurkowski

**Affiliations:** Profesor Sznurkowski Podmiot Leczniczy, 81-346 Gdynia, Poland; jacek.sznurkowski@gumed.edu.pl; Tel.: +48-40-927-05-67

**Keywords:** ovarian cancer, maintenance therapy, bevacizumab, PARPi, guidelines, AGREE

## Abstract

**Simple Summary:**

This study addresses the challenge of selecting the best maintenance therapy for ovarian cancer patients. The two medications under consideration are PARP inhibitors and bevacizumab. The aim is to provide clear guidelines based on scientific evidence that can help oncologists make informed treatment decisions. Upon evaluating various options, it is recommended that bevacizumab be spared for a second line maintenance therapy, while PARP inhibitors should be offered to all advanced ovarian cancer patients who have responded to initial platinum-based chemotherapy. The findings emphasize the importance of ongoing research to identify additional factors that can predict the effectiveness of bevacizumab. By following these guidelines, healthcare professionals can improve the outcomes and quality of life for ovarian cancer patients, contributing to better overall management of the disease and its impact on society.

**Abstract:**

Background: Maintenance therapy with PARP inhibitors and bevacizumab is approved for ovarian cancer treatment in the first and second line settings, but selecting the optimal sequence is challenging due to restrictions on using the same medication twice. This review aims to establish guidelines for ovarian cancer maintenance therapy based on the strength of scientific evidence, the most effective treatment strategy, and the impact on the healthcare system. Methods: Six questions were formulated to evaluate the scientific evidence supporting different maintenance therapy options using the AGREE II guideline evaluation tool. The questions address the acceptability of reusing the same medication, the efficacy of bevacizumab and PARP inhibitors in the first and second line settings, the comparative efficacy of these medications, the potential benefit of combination maintenance therapy, and the economic impact of maintenance therapy. Results: Based on the available evidence, bevacizumab should be preserved for second line maintenance therapy, and maintenance therapy with PARP inhibitors should be offered to all advanced ovarian cancer patients who have responded to first line platinum-based chemotherapy. Additional molecular predictors for bevacizumab efficacy are needed. Conclusions: The presented guidelines offer an evidence-based framework for selecting the most effective maintenance therapy for ovarian cancer patients. Further research is necessary to refine these recommendations and improve outcomes for patients with this disease.

## 1. Introduction

Bevacizumab is a monoclonal antibody that targets and inhibits vascular endothelial growth factor (VEGF), which is a protein that stimulates the growth of new blood vessels. By inhibiting VEGF, bevacizumab reduces the formation of new blood vessels that supply oxygen and nutrients to tumors, thus slowing down their growth and spread [1]. In addition to its anti-angiogenic effects, bevacizumab may also have other mechanisms of action, such as modulation of the immune response and direct effects on cancer cells. However, the exact mechanisms by which bevacizumab exerts its anti-tumor effects are still being studied [2].

PARP inhibitors (PARPis) are a class of drugs that target poly(ADP-ribose) polymerase (PARP), an enzyme involved in DNA repair. When PARP is inhibited, cancer cells with defects in other DNA repair pathways, such as those with BRCA mutations, are unable to repair DNA damage efficiently, leading to cell death [3,4].

PARPis and bevacizumab have been approved as first and second line maintenance therapies for ovarian cancer. However, due to drug registration limits, it is not possible to use bevacizumab after bevacizumab or PARP after PARP. This presents a significant challenge in selecting the best sequence of treatment. It is crucial to consider the varying strength of scientific evidence from published studies on ovarian cancer maintenance therapy in the decision-making process.

The objective of this review is to create a set of guidelines for maintenance therapy based on several factors including the strength of scientific evidence from available trials, the treatment strategy that offers the most benefits to patients, and affordability for the healthcare system. The guidelines were developed according to the standards set by the AGREE II (appraisal of guidelines for research and evaluation) guideline evaluation tool. It comprises 23 items that are categorized into six domains, among which the rigor of development domain holds significant importance. Key points for the rigor of development domain include:Quality Assessment: The guideline should provide comprehensive details on the assessment of study quality and risk of bias for the included studies. This should encompass the tools or criteria used for evaluation;Evidence Synthesis: The guideline should clearly outline the methods employed to synthesize the evidence gathered from relevant studies; andRecommendations Formulation: The guideline should provide a clear description of the process used to formulate recommendations. This should include how the evidence was graded and how the values and preferences of the target population were taken into account.

In my approach, I primarily focus on addressing clinical questions by comparing the strength of scientific evidence derived from available data.

The resulting guidelines aim to provide healthcare professionals with clear and evidence-based recommendations for maintenance therapy in a way that is both effective and cost-efficient.

## 2. Statistical Background

Most clinical trials on maintenance therapy in ovarian cancer provide results for the general population (confirmatory data) as well as post-hoc analyses (exploratory data). However, it is important to consider whether the strength of evidence for confirmatory data is the same as that of exploratory data in randomized clinical trials (RCTs).

Confirmatory data analysis involves testing specific hypotheses or research questions that were formulated prior to conducting the study. The goal of confirmatory analysis is to establish whether the study’s pre-defined hypotheses are supported by the data or not. This approach uses statistical methods that are pre-specified before the start of the study.

Exploratory data analysis, moreover, involves exploring the data without any preconceived notions or hypotheses. This approach is used to generate new hypotheses and insights from the data that were not previously considered [5].

It is important to note that confirmatory analysis is more rigorous and reliable than exploratory analysis, as it follows a pre-specified protocol and has a lower risk of producing false-positive results. However, exploratory analysis can be useful for generating hypotheses and identifying potential trends or relationships in the data that can be tested in future studies [6].

According to the GRADE system, which is widely used to evaluate the strength of evidence for clinical research, the grade of evidence for results of a phase III RCT can range from high to moderate, depending on the study design, methodology, and risk of bias. Conversely, the grade of evidence for exploratory data analysis would usually be lower and can range from low to very low, depending on the specific study design and methodology [7].

In general, results coming from the exploratory data analyses of RCTs are equal in strength to the results of observational studies [8].

In this review, the strength of scientific evidence was defined based on the guidelines for scientific evidence classification set by The Agency for Health Technology Assessment and Tariff System (AOTMiT) [9]. See Table 1.

This classification does not include network meta-analysis (NMA) which is a statistical method that allows for the comparison of multiple interventions and their relative efficacy and safety. NMA is an extension of traditional pairwise meta-analysis that can compare multiple interventions that may not have been directly compared in randomized controlled trials (RCTs). The strength of evidence in NMA is defined by SUCRA (surface under the cumulative ranking) values which depend on the quality of the underlying RCTs and the methods used to conduct the analysis. Strength of evidence of high SUCRA value indicates scientific evidence of IIB [10].

## 3. Questions

To determine whether or not to “Bev” during ovarian cancer maintenance therapy, we have formulated six key questions that need to be considered in order to arrive at an informed decision:

### 3.1. Q1: Is It Acceptable That Bevacizumab and PARPi Cannot Be Administered Again after Previous Treatment with the Same Medication?

Several clinical trials have investigated the use of PARP inhibitors in patients with ovarian cancer who had received prior PARP inhibitor therapy. For example, in a phase II trial of the PARP inhibitor niraparib, patients who had received prior PARP inhibitor therapy had a lower response rate compared to those who had not received prior therapy [11]. Similarly, in a phase II trial of the PARP inhibitor veliparib, patients who had received prior PARP inhibitor therapy had a lower response rate and shorter progression-free survival compared to those who had not received prior therapy [12].

Regarding bevacizumab, there is evidence that its effectiveness may be reduced in patients who have previously received bevacizumab therapy. For example, a phase III trial of bevacizumab in patients with recurrent ovarian cancer (platin sensitive or platin resistant) showed that patients who had previously received bevacizumab had a shorter progression-free survival compared to those who had not received prior therapy [13,14].

In summary, while there is no direct evidence suggesting that PARP inhibitors cannot be used after prior treatment with other PARP inhibitors, there is some evidence suggesting that they may be less effective. Similarly, there is evidence suggesting that the effectiveness of bevacizumab may be reduced in patients who have previously received bevacizumab therapy.

However, the use of these drugs in ovarian cancer treatment is determined on drug registration regulations limiting the usage of either first or next line settings.

### 3.2. Q2: In Which Treatment Line Is Bevacizumab More Efficacious—First or Second?

There was a total of two phase III randomized controlled trials (RCTs) testing the efficacy of bevacizumab in the first line [15,16], and three in the second line [13,14,17], as detailed in Table 2. The efficacy of bevacizumab is summarized in Table 3.

Confirmatory data analyses (ITT population) of GOG218, ICON7, AURELIA, OCEANS, and GOG213 have shown that Bev increases PFS in second line settings only [13,14,15,16,17] [strength of evidence IIA, IIA, IIA, IIA, IIA]. A systemic review with meta-analysis of RCTs indicated not only PFS benefit but also significant OS improvement for second line bevacizumab maintenance [18] [strength of evidence IA].

The increase of PFS observed in the first line for the high-risk group of patients (HR 0.76, 95% CI 0.68–0.84, I2 = 0%) comes from exploratory data [strength of evidence IIIB] and is lower than the increase of PFS in ITT population of second line maintenance therapy (HR 0.53, 95% CI 0.45–0.63, *p* = 0.12, I2 = 54%) [18] [strength of evidence IA] (see Table 3).

It is important that first line maintenance therapy of GOG218 and ICON7 has been provided in patients with primary debulking surgery (PDS). 

Two phase 2 RCTs have investigated the use of bevacizumab in patients undergoing neoadjuvant chemotherapy (NACT), interval debulking surgery (IDS), and adjuvant chemotherapy (ACT). One of these studies [19] did not find a PFS benefit or improvement in the complete cytoreduction rate (CCR), while the other study examined the impact of bevacizumab on CCR only and suggested an improvement in the treatment arm [20]. However, it should be noted that this study did not compare the treatment arm to a control arm, but to a reference rate of 45% taken from another study. Despite this, bevacizumab has started to be widely used as maintenance for patients with NACT. 

In conclusion, the available confirmatory data suggest that bevacizumab is not effective, or if you want to accept exploratory data, is less effective as maintenance therapy in the first line treatment of ovarian cancer compared to the second line setting. 

This should be considered when planning the treatment strategy for newly diagnosed ovarian cancer patients, as almost all patients will eventually die from recurrence [21]. The most important goal in the treatment of recurrent ovarian cancer is prolonging survival time using an effective and well-tolerated strategy. A meta-analysis of confirmatory data from the OCEANS, AURELIA, and GOG 213 trials demonstrates that bevacizumab combined with chemotherapy is currently the best available treatment option for recurrent disease (platin resistance, platin sensitive with/without secondary cytoreduction surgery), improving both PFS (HR 0.53, 95% CI 0.45–0.63, *p* = 0.12, I2 = 54%) and OS (HR 0.87, 95% CI 0.77–0.99, I2 = 0%) [18] [strength of evidence IA]. This could be lost if bevacizumab is used during first line treatment.

### 3.3. Q3: In Which Treatment Line Is PARPi More Efficacious—First or Second?

Based on the confirmatory data from the SOLO1 [22], PRIMA [23] PRIME [24], ATHENA MONO [25], SOLO2 [26], NOVA [27], ARIEL3 [28], and ARIEL4 [29] trials, PARP inhibitors appear to have similar efficacy in both first and second line settings for ovarian cancer patients. 

Characteristics of these RCTs include Table 4.

Therefore, it is reasonable to consider starting PARP inhibitors at once in eligible patients, as they may benefit from the treatment regardless of the treatment line. Additionally, starting with PARP inhibitors in the first line setting may provide more patients with the opportunity to receive this effective treatment earlier in their disease course, potentially improving their overall outcomes.

### 3.4. Q4: Can PARP Inhibitors Be Considered More Effective Than Bevacizumab in the First Line Treatment of Ovarian Cancer?

A recent network meta-analysis (NMA) of RCT compared the effectiveness and safety of bevacizumab and PARPi in women newly diagnosed with ovarian cancer. The findings revealed that for women with specific genetic mutations, such as BRCAm and HRD, PARPi significantly reduced the risk of ovarian cancer progression compared to bevacizumab (PARPi for BRCAm: and HRD: HR 0.47, 95% CI 0.36–0.60; and HR 0.66, 95% CI 0.50–0.87, respectively; Bev for BRCAm and HRD: HR 0.76, 95% CI 0.67–0.87, and HR 0.76, 95% CI 0.66–0.87, respectively).

However, when considering the overall population, there was no significant difference observed in progression-free survival between PARPi and bevacizumab [30] [strength of evidence IIB].

Notably, in the overall population, women with BRCAm and HRD, and treated with PARPi had the highest SUCRA value, suggesting it as a more favorable treatment option for preventing ovarian cancer progression in the first line setting [strength of evidence IIB].

### 3.5. Q5: Can Combination Maintenance Therapy with Both Bevacizumab and PARPi Provide Better Efficacy Than Bevacizumab Alone or Olaparib Alone in the First Line Treatment of Ovarian Cancer?

Unfortunately, there is currently no available data to answer this question as there are no three-arm randomized controlled trials comparing the efficacy of the combination maintenance therapy (bevacizumab + PARP inhibitor) with bevacizumab or PARP inhibitor alone in the first line setting.

The PAOLA-1 trial evaluated the addition of olaparib to standard first line chemotherapy and bevacizumab as maintenance therapy in patients with advanced ovarian cancer who responded to first line treatment. The trial found that the addition of olaparib to bevacizumab significantly improved PFS compared to bevacizumab alone, with a hazard ratio of 0.59 (95% confidence interval [CI] 0.49–0.72) in the overall population (confirmatory data analysis) [31] [strength of evidence IIA].

In pre-specified subgroups, the HRs for PFS with the addition of olaparib were:Homologous recombination deficiency (HRD) positive: HR 0.33 (95% CI 0.25–0.45);BRCA1/2 mutation: HR 0.31 (95% CI 0.20–0.47); andHRD negative: HR 0.74 (95% CI 0.56–0.97). (Exploratory data analysis) [31] [strength of evidence IIIB].

In conclusion, this trial shows that when starting treatment with bevacizumab in the front line setting, adding olaparib to bevacizumab maintenance improves progression-free survival (PFS) in a subset of patients (platinum-sensitive patients only, comprising 80% of the cohort), particularly those with BRCA mutations and homologous recombination deficiency (HRD). However, it should be noted that this was a two-arm study, and it is unclear whether the combination maintenance regimen, bevacizumab alone, or olaparib alone is responsible for the observed effect.

Interestingly, the exploratory data of this study differ from the exploratory data of previous trials, such as ICON7 and GOG218, which suggest that bevacizumab may be more effective in patients with residual disease. In a post-hoc exploratory analysis, it was observed that the PFS benefit observed in the bevacizumab + olaparib arm was higher in patients without residual disease after primary debulking surgery (i.e., the PAOLA low-risk group) compared to those with residual disease (i.e., the PAOLA high-risk group) (HR 0.15 95% CI 0.07–0.30 vs. HR 0.39 95% CI 0.28–0.54, respectively) [strength of evidence IIIB]. 

According to the exploratory data from the PAOLA-1 trial, the combination of bevacizumab and olaparib may be more beneficial for patients without residual disease. However, it is important to consider that these trials involved different patient populations, treatment regimens, and endpoints, which could have contributed to the contrasting results. 

These conflicting results confirm the low-strength of evidence arising from exploratory data in clinical trials. It is important to await further confirmation from additional studies before drawing any definitive conclusions or making treatment decisions based solely on this exploratory data.

### 3.6. Q6: Do Economic Analyses Exist That Can Guide Decision-Making Regarding Maintenance Therapy in Ovarian Cancer?

A recent network meta-analysis (NMA) evaluated the use of different PARPi regimens in BRCA-mutated ovarian cancer patients who were responsive to front line platinum treatment (bevacizumab and olaparib, veliparib and chemotherapy, olaparib alone) or those with platinum-sensitive relapsed cancer (olaparib, rucaparib, niraparib). The paper compared the clinical benefits, toxicity, and net health benefits of various regimens in phase III randomized controlled trials. The study revealed that the current PARPi regimens demonstrated similar clinical benefits, toxicity profiles, and net health benefits in both the upfront (front line) and relapsed settings. However, the addition of bevacizumab to olaparib was found to increase the cost per unit net health benefit compared to olaparib monotherapy. The upfront PARPi regimens were associated with lower toxicity scores compared to the regimens used at relapse. The authors of the paper concluded that combining PARPi with bevacizumab is not recommended in the upfront setting from a cost-effective perspective [32] [strength of evidence IIB].

## 4. Future Bevacizumab Perspectives as Maintenance Therapy

The current decision-making process to use bevacizumab in ovarian cancer is based on clinical features (residual disease or FIGO stage) from the exploratory analyses of GOG218 and ICON7 studies. However, this approach can lead us to overlook the importance of determining the VEGF expression or angiogenic profile of the tumor, similar to how BRCAm or HRD are used to determine eligibility for PARP inhibitors.

A very recent study has validated the angiogenesis score, which could be used to stratify therapy response to tyrosine kinase inhibitors to provide evidence for patient-tailored oncologic therapy in ovarian cancer [33]. This could be easily implemented into clinical practice considering that (mostly) all tumor specimens are investigated for HRD using RNA sequencing. The analysis of such an approach could be simply extended using angiogenic markers. 

By assessing the angiogenic profile of the tumor, we can better identify patients who are more likely to benefit from antiangiogenic therapy, including bevacizumab. Therefore, it is important to consider the tumor’s VEGF expression and angiogenic profile when deciding whether or not to use bevacizumab in ovarian cancer treatment. Considering the cost and potential harm of such therapies, guiding angiogenic treatment reflects an unmet clinical need from a clinical and economic perspective. 

## 5. Conclusions

Based on a comparison of the strength of scientific evidence from available trials, it appears reasonable to consider using PARP inhibitors as first line maintenance therapy for all patients with advanced ovarian cancer who respond to platinum-based chemotherapy. For patients who do not respond to platinum or who experience recurrence during PARP inhibitor maintenance, bevacizumab may be more suitable as a second line option—it improves PFS and OS (confirmatory data). However, more trials are needed to validate the efficacy of bevacizumab in patients with molecularly confirmed angiogenic tumor profiles (bevacizumab predictors), regardless of the treatment line. The final conclusion of the manuscript provides detailed guidelines that can be summarized as follows:
-Low strength of evidence supports the use of bevacizumab (Bev) in the first line treatment. It is recommended to preserve Bev for second line therapy; and-PARP inhibitors (PARPis) should be used as first line maintenance therapy for all patients who respond to platinum-based treatment.

## Figures and Tables

**Table 1 cancers-15-02980-t001:** Grading criteria according to the AOTMiT (The Agency for Health Technology Assessment and Tariff System) guidelines.

Study Type	Grade	Subtype Description
RTC systematic review	IA	Meta-analysis based on RTC systematic review results
	IB	RCT systematic review without meta-analysis
Experimental study	IIA	Well conducted randomised controlled trial, including pragmatic randomised controlled trial
	IIB	Well conducted clinical controlled trial with pseudorandomisation
	IIC	Well conducted clinical controlled trial without randomisation
	IID	One-arm study
Observational study with control group	IIIA	Systematic review of observational studies
	IIIB	Well conducted prospective cohort studies with simultaneous control group
	IIIC	Well conducted prospective cohort studies with historic control group
	IIID	Well conducted retrospective cohort studies with simultaneous control group
	IIIE	Well conducted case-control study (retrospective)
Descriptive study	IVA	Case series—pretest/posttest *
	IVB	Case series—posttest **
	IVC	Other study of a group of patients
	IVD	Case report
Expert opinion	V	Expert opinions based on clinical experience and reports from expert panels

* Pretest/posttest—a study where measurements are taken both before and after the assessed intervention. ** Posttest—a study where measurements are taken only after the intervention.

**Table 2 cancers-15-02980-t002:** Characteristics of five RCTs.

	GOG218 [6]	ICON7 [7]	OCEANS [8]	AURELIA [9]	GOG213 [10]
Primary Endpoint	PFS	PFS	PFS	PFS	OS
Patients enrolled	Stage III (Incompletely resectable) or Stage IV after PDS	Stage I–III or Stage IVor Inoperable Stage III after PDS	Platinum-sensitive recurrent ovarian cancer (recurrence ≥6 months after completing platinum-based therapy)	Platinum-resistant recurrent ovarian cancer that had progressed ≤6 months after completing platinum-based therapy	Platinum-sensitive recurrent ovarian cancer
Sample size	1248	1528	484	361	748
Control arm	Cycles 1–6: C (AUC 6) + P (175 mg/m^2^) + PL, q3w Cycles 7–22: PL, q3w	Cycles 1–6: C (AUC 5 or 6) + P (175 mg/m^2^), q3w	Cycles 1–10: G (1000 mg/m^2^ on days 1 and 8) + C (AUC 4 on day 1) + PL (15 mg/kg on day 1), q3w	Cycles 1-PD: PAC (80 mg/m^2^ days 1, 8, 15, and 22 q4w); or TOP (4 mg/m^2^, days 1, 8, 15 q4w or 1.25 mg/m^2^, days 1–5 q3w); or PLD (40 mg/m^2^ day 1 q4w)	Paclitaxel (175 mg/m^2^) + carboplatin (AUC5) With or without SCS
Experimental arm	Cycles 1–6: C (AUC 6) + P (175 mg/m^2^) + Bev (15 mg/kg), q3w Cycles 7–22: Bev (15 mg/kg), q3w	Cycles 1–6: C (AUC 5 or 6) + P (175 mg/m^2^) + Bev (7.5 mg/kg), q3w Cycles 7–18: Bev (7.5 mg/kg), q3w	Cycles 1–10: G (1000 mg/m^2^ on days 1 and 8) + C (AUC 4 on day 1) + Bev (15 mg/kg on day 1), q3w	Cycles 1-PD: Chemotherapy + Bev (15 mg/kg q3w or 10 mg/kg), q2w	Bev (15 mg/kg) + P (175 mg/m^2^) + Carboplatin (AUC5), followed by Bev maintenance. With or without SCS

PFS, progression-free survival; OS, overall survival; GOG, gynecological oncology group; C, carboplatin; AUC, area under curve; P, paclitaxel; Bev, bevacizumab; PL, placebo; G, gemcitabine; PAC, weekly paclitaxel; TOP, topotecan; PLD, pegylated liposomal doxorubicin; PD, progressive disease, PDS, primary debulking surgery; SCS, secondary cytoreduction surgery.

**Table 3 cancers-15-02980-t003:** Efficacy results of five RCTs on bevacizumab.

References	Line of Treatment	Primary Endpoint	PFS			OS			ORR (%)
			Median (Months)	HR	HR, 95% CI	Median (Months)	HR	HR, 95% CI	
GOG218	1st line	PFS	10.3	0.770	0.681– 0.870	39.3	0.885	0.750– 1.040	NR
			14.1			39.7			NR
ICON7	1st line	PFS	17.5	0.930	0.830– 1.050	58.6	0.990	0.850– 1.140	48.0
			19.9			58.0			67.0
OCEANS	2nd line	PFS	8.4	0.484	0.388– 0.605	32.9	0.952	0.771– 1.176	57.4
			12.4			33.6			78.5
AURELIA	2nd line	PFS	3.4	0.480	0.380– 0.600	13.3	0.850	0.660– 1.080	12.6
			6.7			16.6			30.9
GOG213	2nd line	OS	10.4	0.614	0.522– 0.722	37.3	0.827	0.683– 1.005	NR
			13.8			42.2			NR

PFS, progression-free survival; OS, overall survival; ORR, objective response rate; CI, confidence interval; NR, not reported.

**Table 4 cancers-15-02980-t004:** Efficacy of PARPi RCTs—confirmatory data analyses for the ITT population.

References	Line of Treatment	Patient Characteristics	Primary Endpoint	PFS		
				Median (Months)	HR	HR, 95% CI
SOLO1 olaparib	1st line	FIGO III or IV (17%); BRCA mut-100%; PDS/IDS—63%/35%; R = 0 cm—76%	PFS	13.8	0.33	0.25–0.43
				56		
PRIMA niraparib	1st line	FIGO III or IV (35%); BRCA mut-30%; PDS/IDS—33%/67%; R = 0 cm—excluded	PFS	8.2	0.62	0.50–0.76
				13.8		
PRIME niraparib	1st line	FIGO III or IV (28%); BRCA mut-33%; PDS/IDS—53%/47%; R = 0 cm—75%	PFS	8.3	0.45	0.34–0.60
				24		
ATHENA MONO rucaparib	1st line	FIGO III or IV (24%); BRCA mut-22%; PDS/IDS—49%/51%; R = 0 cm—62%	PFS	9.2	0.52	0.40–0.68
				20.2		
SOLO2 olaparib	2nd line	Relapsed platinum-sensitive FIGO III or IV, BRCA mut—79%	PFS	5.5	0.30	0.22–0.41
				19.1		
NOVA niraparib	>2rd line	Relapsed platinum-sensitive FIGO III or IV, BRCA wt—n = 350	PFS	NR	0.35	0.230–0.532
				NR		
NOVA niraparib	>2rd line	Relapsed platinum-sensitive FIGO III or IV, BRCA mut—n = 203 (100%)	PFS	NR	0.24	0.131–0.441
				NR		
ARIEL3 rucaparib	>2rd line	Relapsed platinum-sensitive FIGO III or IV, BRCA mut—35%;	PFS	5.4	0.36	0.30–0.45
				10.8		
ARIEL4 rucaparib	>2rd line	BRCA mut 100% Relapsed platin sensitive or platin resistant FIGO III or IV	PFS	5.7	0.67	0.52–0.86
				7.4		

PFS, progression-free survival; rate; CI, confidence interval; NR, not reported.

## Data Availability

Not applicable.

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
