# Peer review of "To Bev or Not to Bev during Ovarian Cancer Maintenance Therapy?"

_cancers, 2023, doi:10.3390/cancers15112980_

Round 1
Reviewer 1 Report
In this review paper, by systemic review on the published clinic trials, Sznurkowski aims to establish guidelines for ovarian cancer maintenance therapy using Bev and PARPi. It brought up a couple of major concerns about maintenance therapy, and offered possible explanations through the results from clinic trails. The design of the paper and the findings showed certain novelty and significance. However, there are some flaws and issues that need further clarification.
1. Although we already have some clinical practice guidelines for ovarian cancer to follow worldwide. This paper do have some guidance for concrete problems which could affect clinical decision-making.
2. This study “aims to establish guidelines for ovarian cancer maintenance therapy using Bev and PARPi”. However, it doesn’t seems like a “creation” of “guidelines” as it is for now. What are the details for the “guidelines” they made? Intensive revision is suggested regarding this.
3. It was mentioned in the abstract and introduction that, “The guidelines were developed according to the standards set by AGREE II guideline evaluation tool”. Please include the detail in the section of method.
Author Response
Thank you for reviewing this paper.
- Although we already have some clinical practice guidelines for ovarian cancer to follow worldwide. This paper do have some guidance for concrete problems which could affect clinical decision-making. Answer:
Many of the currently available guidelines are consensus-based and have included newly registered drugs without adequately comparing their clinical benefits, toxicity profiles, and overall net health benefits.
The aim of this review is to establish a set of guidelines for maintenance therapy that takes into account independent factors, rather than relying solely on consensus. These factors include the strength of scientific evidence derived from available trials, the treatment strategy that provides the maximum benefits to patients, and the affordability of the proposed therapy for the healthcare system.
- This study “aims to establish guidelines for ovarian cancer maintenance therapy using Bev and PARPi”. However, it doesn’t seems like a “creation” of “guidelines” as it is for now. What are the details for the “guidelines” they made? Intensive revision is suggested regarding this. Answer:
The final conclusion includes a detailed guide for decision making process but to enhance clarity, a summary will also be provided. See below:
The final conclusion of the manuscript provides detailed guidelines, which can be summarized as follows:
- Low strength of evidance supports the use of Bevacizumab (Bev) in the first-line treatment. It is recommended to preserve Bev for second-line therapy.
- PARP inhibitors (Parpi) should be used as first-line maintenance therapy for all patients who respond to platinum-based treatment.
- It was mentioned in the abstract and introduction that, “The guidelines were developed according to the standards set by AGREE II guideline evaluation tool”. Please include the detail in the section of method. Answer: Details were added into manuscript:
It comprises 23 items that are categorized into six domains, among which the Rigor of Development domain holds significant importance. Key points for the Rigor of Development domain include:
- Quality Assessment: The guideline should provide comprehensive details on the assessment of study quality and risk of bias for the included studies. This should encompass the tools or criteria used for evaluation.
- Evidence Synthesis: The guideline should clearly outline the methods employed to synthesize the evidence gathered from relevant studies.
- Recommendations Formulation: The guideline should provide a clear description of the process used to formulate recommendations. This should include how the evidence was graded and how the values and preferences of the target population were taken into account.
In our approach, we primarily focus on addressing clinical questions by comparing the strength of scientific evidence derived from available data.
Reviewer 2 Report
- Are the drug registration limits of bev after bev or parp after parp country specific?
- The author discussed two small phase 2 trials of PaRP after PaRP and states that the RR and PFS was shorter compared to those that did not have previous PaRP. This needs to be elobarted as to how a 2nd PaRP compared to initial PaRP without a second PaRP. This is a far more important question that if a PaRP for the 1st time in a second line is better than repeat PaRP
- Same point for Bev
- Discussion needs to be expanded on the OS benefit for Bev in 1st line suboptimal and stage 4 cancers.
- The authors state that PFS is better in the 2nd line setting with improved evidence, but are looking at the HR alone. What was the absolute differences in actual time? i.e. months or days
- HRP population the discussion is more complicated. HRD and BRCAm is clearer that PaRP is better in front line. In the suboptimal and stage 4 HRP the discussion needs to be separated and expanded.
- In the cost/economic analysis section the authors should also expand on bioidenticals and how this will decrease costs.
- This is a very interesting discussion of a complex issue with very little data. It is easy to agree with the authors for most of the discussion. The two areas of TRUE conflict, however are:
o 1. Suboptimal debulked or stage 4 BRCA -ve, HRP patients. The PFS for PaRPs is relatively small and potentially a larger benefit and OS benefit to Bev. Having said that the use of PaRP in 2nd line also has difficulties, which the authors may want to expand on.
o 2. Suboptimal debulked or stage 4 BRCA +ve. PAOLA was a poorly conceived trial, but did show us that you can safely combine the two drugs. No one would argue that a PaRP was not necessary, but the addition of Bev remains controversial.
Author Response
Thank You for reviewing this manuscript.
Are the drug registration limits of bev after bev or parp after parp country specific? Answer: Yes, they are.
- The author discussed two small phase 2 trials of PaRP after PaRP and states that the RR and PFS was shorter compared to those that did not have previous PaRP. This needs to be elobarted as to how a 2nd PaRP compared to initial PaRP without a second PaRP. This is a far more important question that if a PaRP for the 1st time in a second line is better than repeat PaRP Answer:
This question was answered: Ther is no difference in efficacy of 1st time PARPi between I and II line. Please read the answer for the clinical question 3
Q3: In which treatment line is PARPi more efficacious - first or second?
- Same point for Bev
Answer:
Please read the answer for the clinical question 2
Q2: In which treatment line is bevacizumab more efficacious - first or second?
- Discussion needs to be expanded on the OS benefit for Bev in 1st line suboptimal and stage 4 cancers.
Answer: There is no OS benefit for BEV (see table 3)
- The authors state that PFS is better in the 2nd line setting with improved evidence, but are looking at the HR alone. What was the absolute differences in actual time? i.e. months or days Answer: HR provides information about the relative treatment effect, while months give a sense of the absolute duration of survival or disease control in particular trials. When comparing the results of drug interventions across different populations or trials, using the relative HR is preferred. It allows for a direct comparison of the treatment effect between different groups or interventions, irrespective of the baseline risk or characteristics of the populations. Relative HR provides a standardized measure of treatment effect that can be more easily compared across studies or populations
- HRP population the discussion is more complicated. HRD and BRCAm is clearer that PaRP is better in front line. In the suboptimal and stage 4 HRP the discussion needs to be separated and expanded.
Answer The data for the mentioned clinical subgroup are exploratory, indicating low evidence. See further answers
- In the cost/economic analysis section the authors should also expand on bioidenticals and how this will decrease costs. Answer: We do not present our own analysis. Instead, we have identified scientific documents that are available and specifically focused on non-generic drugs.
- This is a very interesting discussion of a complex issue with very little data. It is easy to agree with the authors for most of the discussion. The two areas of TRUE conflict, however are:
o 1. Suboptimal debulked or stage 4 BRCA -ve, HRP patients. The PFS for PaRPs is relatively small and potentially a larger benefit and OS benefit to Bev. Having said that the use of PaRP in 2nd line also has difficulties, which the authors may want to expand on. Answer: The data for Bevacizumab (Bev) in the mentioned clinical subgroup are exploratory, indicating low evidence. On the other hand, the data for PARP inhibitors (PARPi) in the same clinical subgroup are confirmatory, indicating high evidence. For a more detailed understanding of the statistical background, please carefully review the paragraph dedicated to the statistical background in the manuscript
- Suboptimal debulked or stage 4 BRCA +ve. PAOLA was a poorly conceived trial, but did show us that you can safely combine the two drugs. No one would argue that a PaRP was not necessary, but the addition of Bev remains controversial. Answer: Three-arm randomized controlled trials comparing the efficacy of combination maintenance therapy (bevacizumab + PARP inhibitor) with bevacizumab or PARP inhibitor alone in the first-line setting are needed to solve this decision problem